# CHINATRAVEL: A REAL-WORLD BENCHMARK FOR LANGUAGE AGENTS IN CHINESE TRAVEL PLANNING

## ABSTRACT

Recent advances in Large Language Models (LLMs), particularly in language reasoning and tool-use capabilities have sparked the rapid development of *Language Agents* to assist humans across various real-world applications. Among these, travel planning stands out as a significant domain, presenting both academic challenges and practical value due to its inherent complexity and real-world relevance. However, existing travel plan benchmarks do not test language agents with human users or their ability to follow customized requirements, both of which are vital for deploying them in real-world applications. In this paper, we propose China-Travel, a new benchmark tailored to authentic Chinese travel requirements, aiming to provide a more realistic evaluation framework for future language agents. We collect the travel requirements through questionnaires and employ an efficient and faithful evaluation process with 46 metrics covering feasibility, constraint satisfaction, and preference comparison. Moreover, we identify three challenges in the real-world deployments of travel planning, including *constraint recognition*, *concept openness*, and *customized preference*. The empirical studies show that even state-of-the-art neural-symbolic agents succeed in 51.3% constraint validation of human queries. Our findings point to the need for methods that can improve the ability of agents to understand diverse intentions or keep track of constraints with emerging concepts from human requirements.

## 1 INTRODUCTION

A long-standing goal in AI is to build planning agents that are reliable and general, able to assist humans in real-world environments. Recently, Large Language Models (LLMs) (Brown et al., 2020; Ouyang et al., 2022; Achiam et al., 2023) have demonstrated remarkable potential in achieving human-level understanding and planning capabilities. This has sparked the rapid development of a field called *Language Agents*, employing LLMs to perceive the surroundings, reason the solutions, and take appropriate actions, ultimately building an autonomous planning agent (Shinn et al., 2024; Yao et al., 2023; Xi et al., 2023). Equipping LLMs born from web-scale corpora, language agents demonstrate a proficient ability to understand general natural language instructions and collect domain-specific information via tools (Yao et al., 2022; Xie et al., 2023; Jimenez et al., 2024). It alleviates the need for intensive domain-specific goal definition and model deployment with traditional rule-based or reinforcement-learning-based agents, showing few-shot generalization across various domains. This presents a solid step toward the goal of building general artificial intelligence.

Travel planning stands out as a significant domain, presenting both academic challenges and practical value due to its inherent complexity and real-world relevance. However, LLMs are still not able to accurately solve complex combinatorial optimization problems and tend to provide infeasible plans in travel planning. In a recently proposed U.S. domestic benchmark TravelPlanner (Xie et al., 2024) with intercity itinerary planning, the advanced LLM, GPT-4, only achieves a success rate of 0.6%. This result is disappointing and might make one pessimistic about the capabilities of Language Agents in travel planning. However, a few months later, Hao et al. (2024) introduced a neural-symbolic solution, which incorporates formal verification tools into language agents and achieved a 97% success rate on the LLM-synthesized from TravelPlanner benchmark. This progress has dual implications. On one hand, it leads to optimism regarding the potential of *Neuro-symbolic Language Agents*. On the other hand, it prompts further inquiry into the practical applicability of these solutions in addressing real-world travel requirements.

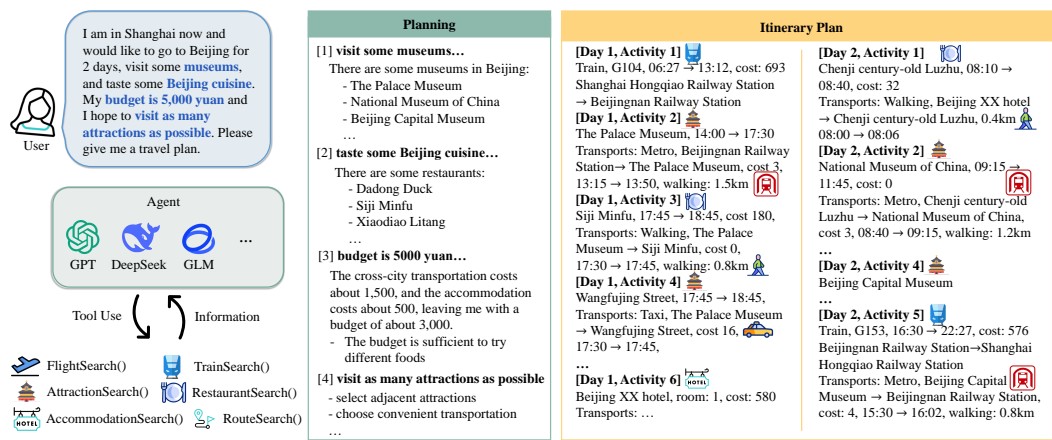

Figure 1: Overview of ChinaTravel. Given a query, language agents employ various search tools to gather information and plan a multi-day multi-POI itinerary. The language agents are expected to provide a feasible and reasonable plan while simultaneously satisfying the hard logical constraints and soft preference requirements. To provide convenience for global researchers, we provide an English translation of the original Chinese information here.

In this work, we introduce ChinaTravel, tailored to authentic Chinese travel requirements, providing a more practical evaluation framework within diverse travel requirements. ChinaTravel concentrates on multi-point-of-interest (multi-POI) itineraries within specified cities (as illustrated in Figure 1), which are in higher demand compared to the intercity itineraries provided by TravelPlanner. China-Travel is built in a modular framework with (1) a rich sandbox environment with Chinese travel information, (2) diverse evaluation metrics covering feasibility, constraint satisfaction, and preference comparison, and (3) realistic travel requirements contain both LLM-synthetic and human question-naire queries. We constructed ChinaTravel in five stages, including manual schema and API design, LLM-assisted generation of data entries, manual quality control, data collection from human users with open requirements, and preference data construction. Our evaluation pipeline automatically verifies the provided plans with the requirements annotations. An additional subset with rich travel preferences is constructed to provide an evaluation for future language agents. Moreover, we identify three challenges in the real-world deployments of travel planning, including constraint recognition, concept openness, and customized preference. The empirical studies show that even state-of-the-art neural-symbolic agents succeed in 51.3% constraint validation of the human queries. Our findings point to the need for methods that can improve the ability of agents to understand diverse intentions or keep track of constraints with emerging requirements from humans.

## 2 RELATED WORK

**Large Language Model based Agents** have demonstrated significant capability in understanding complex instructions and employing domain-specific tools to complete tasks, showcasing their potential in fields such as visual reasoning (Gupta & Kembhavi, 2023), healthcare (Zhang et al., 2023) and robotics (Liu et al., 2024b). This reduces the reliance of previous agents on domain-specific efforts, that is, either mainly following domain-specific rules to plan (rule-based agents, such as DeepBlue (Campbell et al., 2002) and Eliza (Sharma et al., 2017)) or mainly learning from domain-specific data to plan (reinforcement-learning-based agents, such as AlphaGo (Silver et al., 2017) and Atari DQN (Mnih et al., 2013)). While the language agents have shown promising results in some domains, most of their planning scenarios are limited to simple tasks with single objective function and fail in the travel planning benchmark with complex logical constraints on the results.

**Neuro-Symbolic Learning** explores to combine traditional symbolic reasoning with learning to enhance the reliability (Manhaeve et al., 2018; Wang et al., 2019; Dai et al., 2019). In the era of large language models, Pan et al. (2023) presents the LogicLM integrates LLMs with separate symbolic

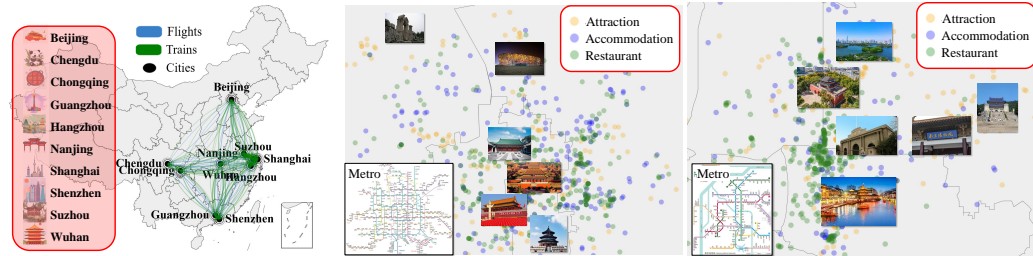

Figure 2: Overview of ChinaTravel sandbox environment. Our sandbox involves travel information from 10 of the most popular cities in China. ChinaTravel provides rich information about the attractions, accommodations, and restaurants that need to be involved in travel. Here is the visualization of information from Beijing and Nanjing.

solvers for various logical reasoning tasks. They first utilize LLMs to translate a natural language problem into a symbolic formulation. Afterward, a deterministic symbolic solver performs inference on the formulated problem to ensure the correctness of the results. Deng et al. (2024) supplement LogicLM with a Self-Refinement Module to enhance the reliability of LLM translation. In the travel planning domain, Hao et al. (2024) presents a framework with a similar pipeline. It first extracts the logical constraints from natural language queries and then formalizes them into SMT code. Thanks to SMT solvers being sound and complete, this neural-symbolic solution guarantees the generated plans are correct and has basically solved the TravelPlanner benchmark (achieved a 97% pass rate).

**Travel Planning** is a time-consuming task even for humans, encompassing travel-related information gathering, POI selection, route mapping, and customization to meet diverse user needs (Halder et al., 2024). Natural languages are one of the most common ways for users to express their travel requirements. However, the ambiguity and complexity of travel requirements make it still challenging for LLMs to generate accurate and reliable travel plans. Xie et al. (2024) presents the TravelPlanner benchmark for cross-city travel planning and reveals the inadequacies of pure-LLM-driven agents. TravelPlanner generates user queries through LLMs and provides a rigorous evaluation mechanism to verify whether the provided plans can meet the logical constraints in the queries. It has become a pivotal benchmark for language agents in real-world travel planning. Tang et al. (2024) study the open-domain urban itinerary planning where a single-day multi-POI plan is required. They integrates spatial optimization with large language models and present a system ITTNERA, to provide customized urban itineraries based on user needs. A concurrent work, TravelAgent (Chen et al., 2024), also considers a multi-day multi-POI travel planning problem for the specified city. It constructs an LLM-powered system to provide personalized plans. However, due to the high cost of collecting and annotating real travel needs, they evaluate the proposed TravelAgent in only 20 queries. This also demonstrates the necessity of introducing a new benchmark for travel planning.

## 3 CHINATRAVEL BENCHMARK

Motivated by the significant travel demand in China, this benchmark offers a sandbox environment for generating multi-day, multi-POI itineraries for specified cities. It includes arrangements for attractions, restaurants, accommodations, and transportation between events, aiming to advance the practice of language agents solutions for real-world travel planning.

ChinaTravel comprises 46 diverse evaluation metrics, including 23 environment constraints, 10 hard logical constraints, and 13 preference requirements, which are summarized in the Table 1. Through manual annotation and formalized code construction, we have built an automated evaluation pipeline for these requirements of given natural language queries, enabling developers to effectively evaluate the capabilities of language agents in addressing real-world challenges.

To evaluate capabilities in real applications, ChinaTravel provides both LLM-synthesized and human queries. We develop pure-LLM-based and neuro-symbolic language agents using the LLM-synthesized queries as a validation set. We then test these agents on human queries, creating an open test environment with real-world dilemmas. The details are provided in the subsection 3.5.

Table 1: Descriptions of evaluation for two benchmarks. Constraints in black are common in both TravelPlanner and ChinaTravel. Metrics in brown are the metrics only in our benchmark.

| Evaluation Metrics | Description |
|---|---|
| *Environment Constraint* | |
| Cross-city Transportation | Available Trains or Airplanes across cities. |
| | Correct information of cost and schedule. |
| Inner-city Transportation | Available Metro, Taxi or Walking between different positions. |
| | Correct information of cost, distance and duration |
| Attractions | Available Attractions in the target city, visiting in their open time. |
| | Attraction choices should not be repeated throughout the trip. |
| | Correct information of cost. |
| Restaurants | Available Restruants in the target city, visiting in their open time. |
| | Restaurant choices should not be repeated throughout the trip. |
| | Breakfast, lunch, and dinner are served at their designated meal times. |
| | Correct information of cost. |
| Accommodation | Available Accommodation in the target city. |
| | Room information to meet headcounts. |
| Time | The given activity events occur in chronological order. |
| Space | Events at different positions should provide transport information. |
| *Logical Constraint* | |
| Transportation | The required type of transportation. |
| Attraction | The required type or specified attractions. |
| Restruants | The required type or specified restruants. |
| Accommodation | The number of rooms and the room type must meet the requirements. |
| | The required features or specified hotels. |
| Budget | The total cost is within required budget. |
| *Preference Requirement* | |
| Transportation | Convenient transportation, less walking distance |
| Attraction | More/less cost on attractions, visit more attractions, |
| | visit more attractions with the required type. |
| Restruants | More/less cost on meals. |
| Accommodation | More/less cost on hotel. |
| Budget | Minimize the total budget. |
| Time | Unhurried itinerary. |
| Space | Schedule the activitiess close to the required position. |

## 3.1 ENVIRONMENT INORMATION

ChinaTravel provides a sandbox with real-world travel information. We collect information from 10 of the most popular cities in China, including Beijing, Chengdu, Chongqing, Guangzhou, Hangzhou, Nanjing, Shanghai, Shenzhen, Suzhou, and Wuhan. There are 720 airplanes and 5770 trains across these cities. Each record contains departure and arrival times from origin to destination, as well as the corresponding ticket prices. We also collect information on 3413 attractions, 4655 restaurants, and 4124 hotels. Each record contains the name, location, opening hours, and the corresponding price per person. Moreover, there are type annotations for these POIs as information to meet user needs. Figure 2 has demonstrated the travel information from Beijing and Nanjing, two of the most popular cities in China. For a more realistic interaction, we simulate the API interface of real market applications to query real-time information. The detailed designs of the sandbox are available in Appendix A. The environmental constraints are designed to ensure the reliability of the results. That is, the POIs visited in the plan must exist in the corresponding city, the transportation methods provided in the plan must be feasible, and the corresponding time information should also be reliable. For example, there should indeed be a subway line that can depart from Beijing Capital International Airport and arrive at the Palace Museum in 80 minutes.

## 3.2 LOGICAL CONSTRAINT

A crucial ability for agents is to effectively satisfy personalized user needs. We extend the logical constraints from TravelPlanner (Xie et al., 2024) to adapt to the multi-POI itinerary planning problem. These user needs are termed logical constraints, which could be defined through logical expressions based on human-defined symbolic concepts. Taking the query in Figure 1 as an example, the user has mentioned "visit the museum", "taste Beijing cuisine", and "budget is 5000 yuan", the provided plan should satisfy the following logical expressions: museum $\in$ attractions_type_visited (plan), Beijing cuisine $\in$ restaurants_type_visited (plan), and cost(plan) $\leq$ 5000, where these symbolic concpets, attractions_type_visited, restaurants_type_visited and cost could be extracted from the formulated plans (as illustrated in Figure 1). ChinaTravel invloves 16 travel-related symoblic concepts to meet the various user needs. We provide a summary and the detailed descriptions of these concepts in Table 1.

## 3.3 PREFERENCE REQUIREMENT

Travel requirements not only include hard logical constraints but also soft preferences. The "soft" means these requirements cannot be defined as constraint validation on discrete symbolic concepts, but rather as quantitative comparisons with the related continuous concepts. This makes the evaluation of preference requirements different from logical constraints. In ChinaTravel, we define 20 concepts for the 13 preferences to provide a ranking-based evaluation. Specifically, we extract relevant concepts from plans generated by different agents, such as the number of attractions visited, walking distance, total cost, etc. We then use these statistics to rank the agents, ultimately providing an automated evaluation mechanism. The detailed concept descriptions are provided in Table 1.

## 3.4 BENCHMARK CONSTRUCTION

ChinaTravel establishes a travel environment in terms of a rich database, API code, and the users' queries with personal requirements. The overall benchmark is created in a five-stage approach with a mix of LLM generation and human survey.

**Stage I: Manual design of database schema and APIs.** We started collecting travel information with the motivation of multi-day multi-POI itinerary planning in four aspects: attractions, accommodation, activities, and transportation. Developers first determine the POI description information that needs to be obtained from the user's perspective, such as cuisine and hotel features. Based on this feature set, we collect public information to construct the database. For the design of APIs, we directly support queries based on the regular expressions from agents, which we hope will promote the use of advanced tools during planning. At the same time, we expect the design of APIs to have similar features and characteristics to existing commercial APIs, enabling our dataset to be applicable to more realistic scenarios.

**Stage II: Automatic data generation with LLMs.** We designed common travel information (origin, destination, days, number of people) and logical constraints based on the nature of travel tasks. To facilitate scalable queries for ChinaTravel, we randomly constructed query skeletons from the aforementioned information and used advanced large language models (e.g. GPT4o) to generate natural language queries from these skeletons. The automatically generated data is categorized into two difficulty levels: `Easy` and `Medium`. In `Easy` level, the logical constraints are straightforward, and the descriptions for the defined concepts in natural language queries align perfectly with these constraints. At the `Medium` level, the natural language expressions of logical constraints are more varied and human-like. For example, the logic 'Beijing cuisine $\in$ restaurants_type_visited(plan)' might correspond to the natural language query: 'I want to try local food in Beijing'. We employ prompt engineering to guide LLMs in modifying the natural language expressions to achieve automated generation.

**Stage III: Manual quality control and automaticed validation.** To ensure data quality, we manually check whether the generated queries conform to symbolic skeletons, and re-calibrate natural language descriptions that contain ambiguities. Additionally, we calibrate the natural language concept descriptions in `Medium` to closely align with human questioning habits. Based on the symbolic

skeletons, we could verify whether the plan can pass the required logical constraints by executing the corresponding Python code. Building on this, we ensure that each query has at least one solution that satisfies the logical constraints by implementing a heuristic search algorithm.

**Stage IV: Open requirements from humans.** After the first round of closed-loop development based on LLM-generated queries, including data generation and annotation, baseline development, and evaluation, we further collected travel requirements from more than 250 humans through questionnaires. Based on a new round of manual quality control on these open requirements, a more challenging set with 154 queries is constructed. These queries even include logical constraints on undefined concepts in the deployment process, such as 'departure time' and 'hot spots', reflecting the real challenges of neural-symbolic systems in travel planning. We carefully annotate the required logical constraints for each query, enabling the automated evaluation of these challenging samples and forming the `Human` level dataset. While we have supported the automated testing of logic constraints with undefined concepts, we hope future researchers avoid making these concepts transparent when utilizing the `Human` set, in order to maintain their openness.

**Stage V: Preference data construction.** Through our investigation of human-annotated queries, we identified that certain human requirements could not be expressed as hard logic constraints, such as "minimize cost" and "maximize convenience in transportation." We classified these as soft preferences of human needs. To better evaluate the performance of these preferences, we distilled and summarized preferences found in `Human` and automatically constructed `Preference` set of 146 samples using the method in stage II. We provided annotations of these preferences for each sample and manually cleaned the data to facilitate further research.

To promote global research on travel planning, we provide the English version of all the queries in the ChinaTravel Benchmark. Despite this, we recommend that researchers mainly use the Chinese version, which can reflect the needs of native speakers more accurately. As discussed above, this raises the critical challenges for Language Agents in travel planning.

### 3.5 KEY CHARACTERISTICS

**Arbitrary description for the defined concepts.** The success of neural-symbolic solutions relies on accurate translation from natural languages to human-defined concepts. We find that even for advanced LLMs, it is still challenging to understand the diverse descriptions of human queries. The variability in human language, including ambiguous phrasing, context-dependent meanings, and open-ended expressions, makes it difficult for LLMs to map these descriptions to predefined concepts. This gap often results in failures when the models attempt to reconcile flexible human input with pre-defined symbolic structures, hindering their performance in tasks requiring precise constraint recognition and adherence to user preferences.

**Emergence of the undefined concepts.** In real-world applications, language agents will encounter symbolic concepts that were not predefined during development, making it challenging to satisfy the related constraints. Real-world concepts are dynamic and consistently evolving, making it unrealistic to rely on a closed concept library to handle open-world demands. Therefore, neural-symbolic language agents must learn to recognize and adapt to new concepts as they emerge in an open-world environment, expanding their symbolic knowledge base to ensure scalability and robustness.

**Diverse preference requirements.** Real requirements also involve customized preferences which are challenging for language agents. On the one hand, due to the diversity of human expressions, LLMs often struggle to accurately interpret these preferences. For instance, a query like 'prefer not to be under the sun' implies a preference for 'more indoor attractions' necessitating robust intent analysis and a deep understanding of user behavior patterns. Currently, most methods rely on general-purpose models, such as GPT-4, which may lack the specialized capabilities required for this task. On the other hand, even if the LLM can accurately identify human preferences, the symbolic search component lacks effective techniques for efficient searching. This is because integrating preferences with logical constraints transforms the problem into a complex multi-objective mixed discrete constraint optimization problem. Current SMT-based methods and heuristic search techniques often fail to find satisfactory solutions within a limited time frame.

## 4 EMPIRICAL STUDY

We evaluate the performance of both pure-LLM-based and neural-symbolic solutions on the China-Travel benchmark. Regarding the former, we primarily tested the well-known method, ReAct (Yao et al., 2023), and its Act-only ablation, where the model is instructed to zero-shot generate "Thought: {some reasoning}, Action: {some formatted action}" or only the action part. Regarding the latter, we follow the neural-symbolic pipelines from (Hao et al., 2024) but replace the SMT solver with a step-by-step depth-first search to adapt to the multi-day multi-POI itinerary. The details will be provided in the subsection 4.1. As for LLMs, we choose the DeepSeek-V2.5 (Liu et al., 2024a) and GLM-4PLUS, which possess advanced Chinese language capabilities, and the GPT-4 as the engine of the language agents. We do not include the given their performance close to ReAct in the TravelPlan benchmark (Xie et al., 2024), the potential benefits of these methods may be limited.

### 4.1 NEURAL-SYMBOLIC SOLUTIONS

Based on the success of the neural-symbolic solution in the TravelPlan benchmark, we adapt the two-stage SMT-based solution to our benchmark, which we call **NeSy Planning**. Following the (Hao et al., 2024), we first extract the logical constraints from the natural language. Based on the extracted constraints, we present a step-by-step plan generation process with depth-first search, that is, mimicking human travel planning by arranging the next activity one by one. Specifically, we first generate the next activity type based on the current plan, and then recursively generate the next activity until the goal is reached. The generated plan is then used to solve the problem.

---

**Algorithm 1** Depth-First Greedy Search

---

**Require:** Constraints $C$, current plan $p$,
  **if** the least activity is an intercity-transport from destination to origin **then**
    **return** ConstraintValidation(p, C), p    ▷ The plan $p$ is finished, return the validation result.
  **end if**
  type = GetNextActivityType(p)    ▷ Select the next type of activities, e.g. lunch, attraction.
  candidates = ToolUse(type)    ▷ Collect the corresponding information for the activity type
  scores = RuleScore(candidates, p, C)    ▷ Score candidates through constraints C.
  **for** activity in candidates **do**
    p.push(activity)    ▷ Perform a greedy search with priority ranking.
    flag, p = Depth-FirstGreedySearch(C, p)
    **if** flag **then**
      **return** True, p    ▷ Return the solution $p$ if the validation is passed.
    **end if**
    p.pop(activity)
  **end for**
  **return** False, p    ▷ Fail to find a solution with the given conditions.

---

For the first step, we follow the (Hao et al., 2024) to implement the translation from natural languages to logical constraints through prompting. The detailed prompts are provided in the Appendix B. In the second step, we define the rule-based activity selection and score function. For example, if the current time is in the [10:30, 12:30] and there is no scheduled lunch in the current plan, then the agent should find a restaurant to have lunch at this time. If the current time is after 22:00 and there are no open-time attractions nearby, the agent should choose to return to the hotel. For the score function, we select the restaurants that satisfy the required cuisine and sort the candidates by the price if there a budget constraints in the constraints $C$. These ranking functions will help us to find a feasible solution as soon as possible. In ChinaTravel, the duration arrangement of activities is continuous and difficult to enumerate and search. We pre-define a meal or a visit to an attraction as 90 minutes, and when there are less than 90 minutes until closing time, the event continues until the closing time. Given these designs, we adapt the neural-symbolic solution into a multi-POI planning problem and evaluate it in the ChinaTravel benchmark.

Table 2: Main results of different LLMs and planning strategies on the ChinaTravel benchmark. LLMs: 🐋: DeepSeek-V2.5, 🌀: GPT-4o-2024-08-06, 🔵: GLM-4PLUS.

| | LLMs | Delivery Rate | Environmental Pass Rate | | Logical Pass Rate | | Final Pass Rate |
|---|---|---|---|---|---|---|---|
| | | | Micro | Macro | Micro | Macro | |
| **Easy (#303)** | | | | | | | |
| Act | 🐋 | 87.1 | 40.7 | 0.33 | 71.0 | 37.0 | 0 |
| | 🌀 | 98.4 | 60.6 | 0 | 85.7 | 44.6 | 0 |
| ReAct | 🔵 | 86.5 | 32.2 | 0 | 58.4 | 18.5 | 0 |
| | 🐋 | 60.4 | 28.1 | 0 | 39.3 | 17.2 | 0 |
| | 🌀 | **99.3** | 42.0 | 0 | 73.8 | 30.4 | 0 |
| ReAct (one-shot) | 🐋 | 92.0 | 62.4 | 9.24 | 85.8 | 62.1 | 7.26 |
| | 🌀 | **99.3** | 61.4 | 0.33 | 93.4 | 72.0 | 0 |
| NeSy Planning | 🔵 | 90.4 | 90.4 | 90.4 | 88.3 | 89.8 | 89.8 |
| | 🐋 | 99.0 | **99.0** | **98.7** | **99.0** | **98.0** | **97.7** |
| | 🌀 | 97.4 | 97.4 | 97.4 | 96.8 | 96.4 | 96.4 |
| **Medium (#180)** | | | | | | | |
| Act | 🐋 | 81.1 | 31.0 | 0 | 64.5 | 43.3 | 0 |
| | 🌀 | 98.9 | 51.9 | 0 | **94.4** | 81.7 | 0 |
| ReAct | 🔵 | 74.4 | 19.1 | 0 | 41.2 | 14.4 | 0 |
| | 🐋 | 58.3 | 22.5 | 1.11 | 31.8 | 13.3 | 0.55 |
| | 🌀 | 98.9 | 33.6 | 0 | 61.1 | 22.8 | 0 |
| ReAct (one-shot) | 🐋 | 83.9 | 49.0 | 2.78 | 75.3 | 54.4 | 2.78 |
| | 🌀 | **100** | 53.4 | 0 | 93.9 | **77.2** | 0 |
| NeSy Planning | 🔵 | 90.0 | 90.0 | 90.0 | 80.9 | 57.8 | **57.8** |
| | 🐋 | 90.6 | 90.5 | 90.0 | 80.8 | 55.6 | 55.6 |
| | 🌀 | 90.6 | **90.6** | **90.6** | 81.3 | 57.8 | **57.8** |
| **Human (#154)** | | | | | | | |
| Act | 🐋 | 75.3 | 26.4 | 0 | 55.0 | 29.9 | 0 |
| | 🌀 | 98.7 | 50.8 | 0 | **80.0** | **54.6** | 0 |
| ReAct | 🔵 | 55.2 | 13.6 | 0 | 33.5 | 16.2 | 0 |
| | 🐋 | 48.7 | 16.6 | 0.65 | 33.6 | 15.0 | 0 |
| | 🌀 | **100** | 34.5 | 0 | 71.3 | 31.2 | 0 |
| ReAct (one-shot) | 🐋 | 79.2 | 41.8 | 2.60 | 64.2 | 42.2 | 2.60 |
| | 🌀 | 70.8 | 37.4 | 0 | 62.7 | 44.8 | 0 |
| NeSy Planning | 🔵 | 62.3 | 62.2 | 61.0 | 49.6 | 42.2 | 41.6 |
| | 🐋 | 55.8 | 55.4 | 52.0 | 45.6 | 37.7 | 35.7 |
| | 🌀 | 79.2 | **78.9** | **77.3** | 62.9 | 51.3 | **51.3** |

## 4.2 MAIN RESULTS

We provide the main results in Table 2. For Easy set, we observe that while most models exhibit a high delivery rate using Act and React (Yao et al., 2023) methods, they perform poorly in constraint satisfaction. Given that the logical constraints in this set are relatively simple (e.g., mostly only involving the number of people and travel days), these methods achieve a favorable logical pass rate. Unlike TravelPlanner (Xie et al., 2024), our task involves multi-day multi-POI scenarios, where satisfying environmental constraints becomes more challenging as the number of POIs increases. Consequently, purely LLM-based methods tend to fail in the environmental pass rate met-

ric, thus resulting in a low final pass rate, with many models failing entirely. We find that the need to document transportation details between large number of POIs often lead to a high frequency of hallucinations in LLMs. Specifically, these models frequently invent transportation information rather than providing the requested result from APIs in the final plan. Our attempts to address this issue through prompt engineering alone have proven insufficient. Notably, Deepseek-V2.5 (Liu et al., 2024a) achieves a 7.26% pass rate in ReAct due to its strong capability in following Chinese instructions. In this set, NeSy unsurprisingly achieved the best results, with the final pass rate approaching 100%. This aligns with the observations in the SMT-based method (Hao et al., 2024), which demonstrates that when LLMs successfully translate natural language into logical constraints, symbolic search can resolve many issues related to constraint satisfaction.

For `Medium` set, we observed that the performance of Act and React shows little difference compared to the `Easy` set. However, the NeSy planning method has a significant performance decline. This is attributed to arbitrary descriptions of defined concepts in the set, which hinder the LLM's ability to accurately translate natural language into logic constraints. This performance decrease aligns with our expectations, indicating that the NeSy planning approach remains insufficient for addressing more complex tasks.

For `Human` set, almost all the methods' performance declines. Since these queries are crafted by humans, they more closely resemble real-world scenarios, presenting a greater challenge for LLMs. Furthermore, the open-ended nature of human queries introduces undefined concepts, which also results in suboptimal performance for the Nesy planning We conduct a detailed analysis of the `Human` results, and manually calculate the error rate distribution of the NeSy planning method across all models. We categorize the errors into five main types: *Missing constraints error*: indicates a failure to translate appropriate logical constraints. *Parsing error*: occurs when

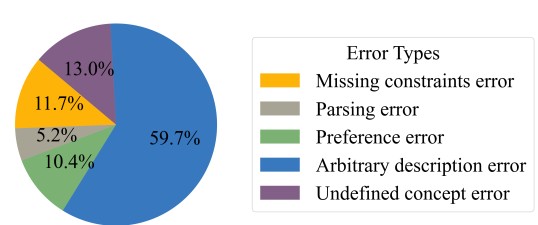

Figure 3: Error distribution for NeSy planning on `Human` set, categorized into five distinct types.

LLMs fail to generate logical constraints in the correct format. *Preference error*: happens when the model mistakenly interprets human preferences as logical constraints. *Arbitrary description error*: arises when the LLMs cannot accurately map human descriptions to well-defined concepts. *Undefined Concept Error*: occurs when an undefined concept prevents the model from converting it into suitable logical constraints. The statistical results of the error distribution are shown in Figure 3. It can be observed that the *Arbitrary Description Error* accounts for the highest proportion at 59.7%, followed by the *Undefined Concept Error*. This indicates that these two issues are the main reasons for the poor performance of the current NeSy planning method on `Human` set. These align with the two key challenges of the NeSy methods proposed in this paper.

## 4.3 CASE STUDY

**Arbitrary description for the defined concepts.** We present two examples of arbitrary descriptions. As shown in Figure 4 (1), a user intends to visit Disneyland. Therefore, Disneyland should be included in the POIs we need to access. However, in the database, Disneyland is listed under its formal name, 'Shanghai Disney Resort'. The issue arises because LLMs cannot access the entire database, leading to errors when translating natural language into symbolic constraints. In the second example, the user wishes to try local cuisine. LLMs extract the term 'local cuisine' as a string, overlooking the intermediate logical relationship that, since the destination is Chengdu, it should specifically refer to 'Sichuan cuisine' which is available in the database.

**Emergence of the undefined concepts.** Two examples of concepts are provided on the Figure 4(2). Although the concepts define that train start and end times should align with the travel information in the database, users often request additional specific time constraints. A more complex example is when, despite having a defined concept for budget, users introduce more intricate constraints, such as excluding airfare from the overall budget. These challenges highlight the cur-

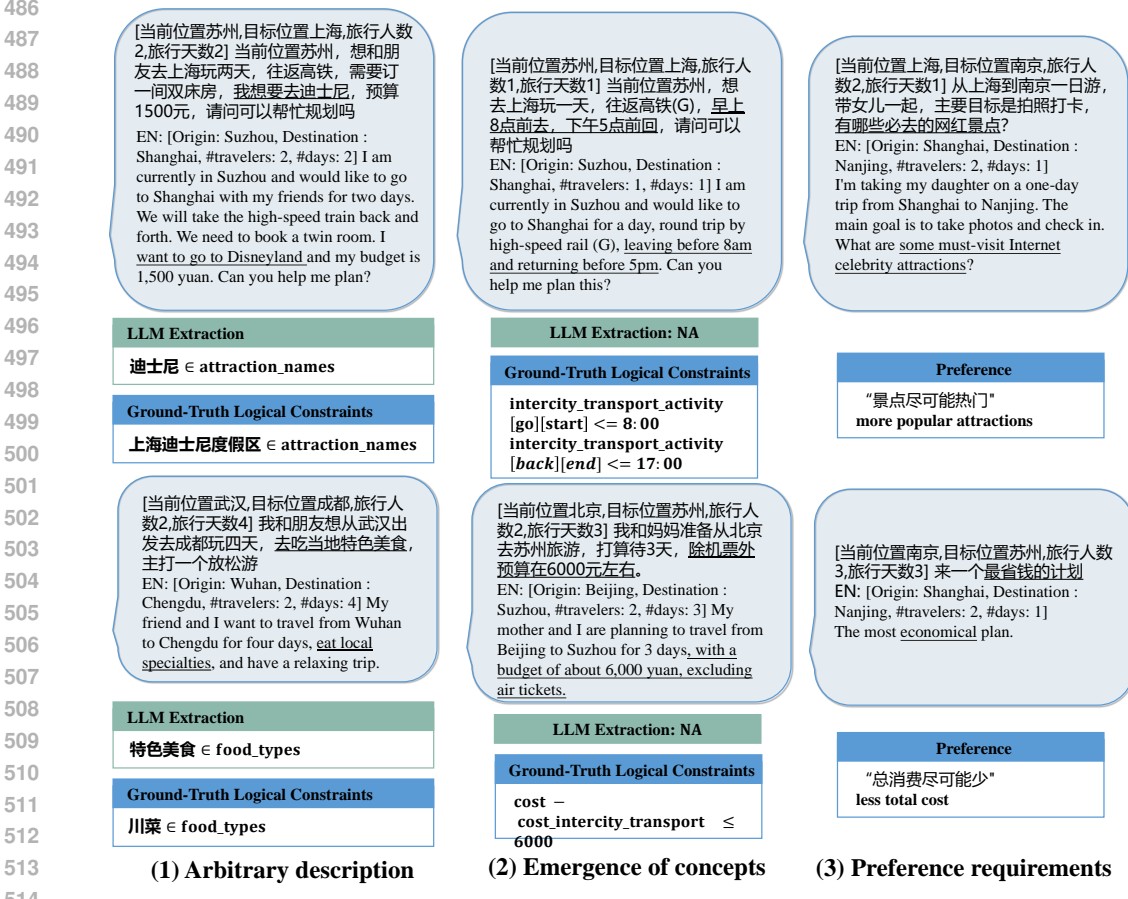

Figure 4: Case study of challenges in real-world travel planning

rent limitations of LLMs and neural-symbolic solutions in translating such emerging constraints and resolving satisfiability issues through symbolic systems.

**Preference Cases.** We present two examples to show how preferences in our benchmark. As shown in Figure 4 (3), a user intends to visit some must-visit attractions. This reflects a user's preference for visiting more popular attractions. Another example is the user's desire for the most economical plan, indicating a preference for lower total cost. These preferences involve undefined concepts, such as the popularity tag of attractions, and require LLMs to have a sufficient understanding of human intentions and a good analysis of behavior patterns. The presence of preferences adds complexity to tasks due to their potential interactions. For instance, there is an inherent conflict between the preference to reduce overall cost and the desire for an enhanced travel experience.

## 5 CONCLUSION

In this paper, we introduced ChinaTravel, a benchmark specifically designed to evaluate language agents in the domain of travel planning, with a focus on authentic Chinese travel requirements. We addressed the limitations of existing benchmarks by incorporating human users and their customized requirements, which are essential for real-world applications. ChinaTravel provides a realistic evaluation framework with diverse metrics covering feasibility, constraint satisfaction, and preference comparison. By addressing the challenges identified in the benchmark, we can pave the way for the deployment of language agents that better meet the customized requirements of users and provide reliable and satisfactory travel planning experiences.

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

Table 3: Database schema.

| POI type | feature list | API |
|---|---|---|
| Attractions | Name, Lat, Lon, Price, Type OpenTime, CloseTime, MinTime, MaxTime | attractions_keys(city) attractions_select(city, key, func) attractions_id_is_open(city, id, time) attractions_nearby(city, point, topk, dist) attractions_types(city) |
| Accommodations | Name, Lat, Lon, Price, NumBed, | accommodations_keys(city) aaccommodations_select(city, key, func) accommodations_nearby(city, point, topk, dist) |
| Restaurants | Name, Lat, Lon, Price, CuisineName, OpenTime, CloseTime, RecommendedFood | restaurants_keys(city) restaurants_select(city, key, func) restaurants_id_is_open(city, id, time) restaurants_nearby(city, point, topk, dist) restaurants_cuisine(city) restaurants_restaurants_with_recommended_food (city, food) |
| Transport | - | goto(city, start, end, start_time, method) intercity_transport_select (start_city, end_city, intercity_type, earliest_leave_time) |
| NoteBook | - | notedown(description, content) |
| Env | - | planner(query) next_page() |

## A  TRAVEL INFORMATION

## B  PROMPTS

```
Act:

PROMPT = """
Collect information for a query plan using interleaving '
    Action', and 'Observation' steps. Ensure you gather valid
     information related to transportation(including inter
    and inner city), dining, attractions, and accommodation.
     All information including time, cost, location and others
     must be written in notebook, which will then be input
    into the Planner tool. Note that the nested use of tools
    is not allowed. 'Action' can have 19 different types:

city_list = ["Shanghai", "Beijing", "Shenzhen", "Guangzhou","
    Chongqing", "Suzhou", "Chengdu", "Hangzhou", "Wuhan", "
    Nanjing"]

(1) attractions_keys(city: str)
Description: Returns a list of (key, type) pairs of the
    attractions data.
Parameters:
city: The city name.
(2) attractions_select(city: str, key: str, func: Callable):
Description: Returns a DataFrame with data filtered by the
    specified key with the specified function.
Parameters:
city: The city name.
key: The key column to filter, only one key can be used.
func: The lambda function applied to the key column, must
    return a boolean value. Only apply to one key.
(3) attractions_id_is_open(city: str, id: int, time: str):
Description: Returns whether the attraction with the
    specified ID is open at the specified time.
Parameters:
city: The city name.
id: The ID of the attraction.
time: The time to check, in the format 'HH:MM'.
(4) attractions_nearby(city: str, point: str, topk: int, dist
    : float = 2):
Description: Returns the top K attractions within the
    specified distance of the location.
Parameters:
city: The city name.
point: The name of the location.
topk: The number of attractions to return.
dist: The maximum distance from the location, default is 2.
(5) attractions_types(city: str):
Description: Returns a list of unique attraction types.
Parameters:
city: The city name.

(6) accommodations_keys(city: str):
Description: Returns a list of (key, type) pairs of the
    accommodations data.
Parameters:
city: The city name.
```

```
(7) accommodations_select(city: str, key: str, func: Callable
    ):
Description: Returns a DataFrame with data filtered by the
    specified key with the specified function.
Parameters:
city: The city name.
key: The key column to filter, only one key can be used.
func: The lambda function applied to the key column, must
    return a boolean value. Only apply to one key.
(8) accommodations_nearby(city: str, point: str, topk: int,
    dist: float = 5):
Description: Returns the top K accommodations within the
    specified distance of the location.
Parameters:
city: The city name.
point: The name of the location.
topk: The number of accommodations to return.
dist: The maximum distance from the location, default is 5.

(9) restaurants_keys(city: str):
Description: Returns a list of (key, type) pairs of the
    restaurants data.
Parameters:
city: The city name.
(10) restaurants_select(city: str, key: str, func: Callable):
Description: Returns a DataFrame with data filtered by the
    specified key with the specified function.
city: The city name.
key: The key column to filter, only one key can be used.
func: The lambda function applied to the key column, must
    return a boolean value. Only apply to one key.
(11) restaurants_id_is_open(city: str, id: int, time: str):
Description: Returns whether the restaurant with the
    specified ID is open at the specified time and day.
Parameters:
city: The city name.
id: The ID of the restaurant.
time: The time to check, in the format 'HH:MM'.
(12) restaurants_nearby(city: str, point: str, topk: int,
    dist: float = 2):
Description: Returns the top K restaurants within the
    specified distance of the location.
Parameters:
city: The city name.
point: The name of the location.
topk: The number of restaurants to return.
dist: The maximum distance from the location, default is 2.
(13) restaurants_restaurants_with_recommended_food(city: str,
     food: str):
Description: Returns all restaurants with the specified food
    in their recommended dishes.
Parameters:
city: The city name.
food: The food to search for.
(14) restaurants_cuisine(city: str):
Description: Returns a list of unique restaurant cuisines.
Parameters:
city: The city name.
```

```
(15) goto(city: str, start: str, end: str, start_time: str,
    method: str):
Description: Returns a list of transportation options between
    two locations.
Parameters:
city: The city name.
start: The start point's name. Must be a location name and
    match the data exactly.
end: The end point's name. Must be a location name and match
    the data exactly.
start_time: The departure time in the format 'HH:MM'.
method: The mode of transportation, must in ['walk', 'taxi',
    'metro'].

(16) notedown(description: str, content: str):
Description: Writes the specified content to the notebook.
Parameters:
description: The description of the content.
content: The content to write.

(17) planner(query: str):
Description: Generates a plan based on the notebook content
    and query.
Parameters:
query: The query to generate a plan for. Don't worry about
    the notebook content, the planner will read it
    automatically.

(18) intercity_transport_select(start_city: str, end_city:
    str, intercity_type: str):
Description: get the intercity transportation information
    between two cities. You need to call this function at
    least twice to get the transportation information between
     two locations for going and returning.
Parameters:
start_city: The start city name.
end_city: The end city name.
intercity_type: The type of intercity transportation, must in
     ['train', 'airplane'].

(19) next_page():
Description: Get the next page of the latest Result history
    if it exists. Because of the length limited, all returned
     DataFrame information is split into 10 rows per page.
    You can use this function to get the next page of the
    Result history. Only DataFrame information can be split
    into multiple pages. The function should not be used too
    often, otherwise, you will soon run out of steps.
Parameters:
None

Your action will be executed in the following format: action,
     so any additional text like 'Action: ' is not allowed
    and just one line is allowed for each action.

You must finish your response within 75 steps including plan.
```

```
Select the transportation, dining, attractions, and
    accommodation information you need to plan your trip and
    write them in the notebook. Not EVERYTHING is needed,
    only what you need to plan the trip. For example, when
    you get ten or more accommodations, you only need to note
     down the information of the accommodation you want to
    stay in, usually one, and note it down in the notebook.
    You must not note down all the accommodations information
    . And usually, 2-4 attractions are enough for one day.

What you note down in the notebook should be a plan or plans
    for days. May be notedown(description = "Day 1(Day 1
    morning is also acceptable)", content = "At 8:00, have
    breakfast at hotel A, then go to attraction B, using
    metro(together with the cost, time, stations and other
    information). Attracion B will cost xxx yuan and xxx
    hours. Then go to restaurant C for lunch, using taxi(
    together with the cost, time, distance and other
    information). Restaurant C will cost xxx yuan.(another
    attraction is possile too as long as there is enough
    time and budget). Then... ###More details here###.")

### EXAMPLE ###

Action[1]: intercity_transport_select(start_city='Beijing',
    end_city='Nanjing', intercity_type='train')
Observation[1]:
Results[1]:
[MASKED]
Action[2]: intercity_transport_select(start_city='Beijing',
    end_city='Nanjing', intercity_type='airplane')
Observation[2]: Please note down what is useful using
    notedown method.
Results[2]:
[MASKED]
Action[3]: intercity_transport_select(start_city='Nanjing',
    end_city='Beijing', intercity_type='airplane')
Observation[3]:
Results[3]:
[MASKED]
Action[4]: notedown(description='Round trip between Beijing
    and Nanjing', content='Heading to Nanjing on flight '
    FL154' from 'Beijing Capital International Airport' to '
    Nanjing Lukou International Airport' at '07:40' arriving
    at '08:47'. The ticket price is 427.98. Returning to
    Beijing on flight 'FL657' from 'Nanjing Lukou
    International Airport' to 'Beijing Daxing International
    Airport' at '18:02' arriving at '19:09'. The ticket price
     is 412.06. Considering that the user is traveling with a
     companion, the round-trip cost between Nanjing and
    Beijing is 1680.08, leaving 2319.92 for planning
    activities, accommodation, and dining within Nanjing.')
Observation[4]:
Results[4]:
NoteBook updated.
Action[5]: attractions_keys(city='Nanjing')
```

```
Observation[5]:
Results[5]:
[MASKED]
Action[6]: attractions_select(city='Nanjing', key='type',
    func=lambda x: True)
Observation[6]:
Results[6]:
[MASKED]
Action[7]: goto(city='Nanjing', start='Nanjing Railay Station
    ', end='Confucius Temple', start_time='08:00', method='
    metro')
Observation[7]:
Results[7]:
[MASKED]

...... // More actions and observations

Action[X]: notedown(description='Day 1', content='At 8:00,
    have breakfast at hotel A, then go to attraction B, using
     metro(together with the cost, time, stations and other
    information). Attracion B will cost xxx yuan and xxx
    hours. Then go to restaurant C for lunch, using taxi(
    together with the cost, time, distance and other
    information). Restaurant C will cost xxx yuan.(another
    attraction is possiple too as long as there is enough
    time and budget). Then... ###More details here###.')

...... // More actions and observations

Action[n]:planner(query="My friend and I plan to visit
    Nanjing for three days with a budget of 4000 yuan. We
    prefer to use the subway as much as possible and enjoy
    Jiangsu and Zhejiang cuisine. Please provide a travel
    plan.")

### EXAMPLE END ###

Do not forget to note down the ###transportation information
    between locations### before planning. Intercity
    transportation information should be noted down before
    planning too.

You need to plan for each day in detail. If only one day is
    planned, accommodation is not needed. If more than one
    day is planned, accommodation is necessary. Nights in
    accommodations should be days-1. For example, if you plan
     for 3 days, you need to note down 2 nights in
    accommodations.

!!!Don't call next_page() too often, only when necessary.!!!
    Once you get the suitable information, you must !!!STOP
    !!! using this function. !!!
```

```
Pay attention to function names and parameters, and the
    format of the data. You must use the correct function
    names and parameters to get the data you need. If you use
     the wrong function names or parameters, you will not get
     the correct data.!!!

It is strictly forbidden to use the next_page() too often!
Remember to note down all information you need in the
    notebook before planning.
"""
```

React:

```
PROMPT = """
Collect information for a query plan using interleaving '
    Thought', 'Action', and 'Observation' steps. Ensure you
    gather valid information related to transportation,
    dining, attractions, and accommodation. All information
    including time, cost, location and others must be written
     in notebook, which will then be input into the Planner
    tool. Note that transportation bwteen locations must be
    written in notebook before planning. Note that the nested
     use of tools is not allowed, 'Thought' can reason about
    the current situation, and 'Action' can have 19 different
     types:

city_list = ["Shanghai", "Beijing", "Shenzhen", "Guangzhou",
    "Chongqing", "Suzhou", "Chengdu", "Hangzhou", "Wuhan", "
    Nanjing"]. All the cities name you use must be in this
    list.

(1) attractions_keys(city: str)
Description: Returns a list of (key, type) pairs of the
    attractions data.
Parameters:
city: The city name.
(2) attractions_select(city: str, key: str = "", func:
    Callable = lambda x: True):
Description: Returns a DataFrame with data filtered by the
    specified key with the specified function.
Parameters:
city: The city name.
key: The key column to filter, only one key can be used. If
    not specified, return all data.
func: The lambda function applied to the key column, must
    return a boolean value. Only apply to one key. If not
    specified, return all data.
(3) attractions_id_is_open(city: str, id: int, time: str):
Description: Returns whether the attraction with the
    specified ID is open at the specified time.
Parameters:
city: The city name.
id: The ID of the attraction.
time: The time to check, in the format 'HH:MM'.
(4) attractions_nearby(city: str, point: str, topk: int, dist
    : float = 2):
Description: Returns the top K attractions within the
    specified distance of the location.
Parameters:
city: The city name.
point: The name of the location.
topk: The number of attractions to return.
dist: The maximum distance from the location, default is 2.
```

```
(5) attractions_types(city: str):
Description: Returns a list of unique attraction types.
Parameters:
city: The city name.

(6) accommodations_keys(city: str):
Description: Returns a list of (key, type) pairs of the
    accommodations data.
Parameters:
city: The city name.
(7) accommodations_select(city: str, key: str = "", func:
    Callable = lambda x: True):
Description: Returns a DataFrame with data filtered by the
    specified key with the specified function.
Parameters:
city: The city name.
key: The key column to filter, only one key can be used. If
    not specified, return all data.
func: The lambda function applied to the key column, must
    return a boolean value. Only apply to one key. If not
    specified, return all data.
(8) accommodations_nearby(city: str, point: str, topk: int,
    dist: float = 5):
Description: Returns the top K accommodations within the
    specified distance of the location.
Parameters:
city: The city name.
point: The name of the location.
topk: The number of accommodations to return.
dist: The maximum distance from the location, default is 5.

(9) restaurants_keys(city: str):
Description: Returns a list of (key, type) pairs of the
    restaurants data.
Parameters:
city: The city name.
(10) restaurants_select(city: str, key: str = "", func:
    Callable = lambda x: True):
Description: Returns a DataFrame with data filtered by the
    specified key with the specified function.
city: The city name.
key: The key column to filter, only one key can be used. If
    not specified, return all data.
func: The lambda function applied to the key column, must
    return a boolean value. Only apply to one key. If not
    specified, return all data.
(11) restaurants_id_is_open(city: str, id: int, time: str):
Description: Returns whether the restaurant with the
    specified ID is open at the specified time and day.
Parameters:
city: The city name.
id: The ID of the restaurant.
time: The time to check, in the format 'HH:MM'.
(12) restaurants_nearby(city: str, point: str, topk: int,
    dist: float = 2):
Description: Returns the top K restaurants within the
    specified distance of the location.
Parameters:
```

```
city: The city name.
point: The name of the location.
topk: The number of restaurants to return.
dist: The maximum distance from the location, default is 2.
  (13) restaurants_restaurants_with_recommended_food(city: str
      , food: str):
Description: Returns all restaurants with the specified food
    in their recommended dishes.
Parameters:
city: The city name.
food: The food to search for.
(14) restaurants_cuisine(city: str):
Description: Returns a list of unique restaurant cuisines.
Parameters:
city: The city name.

(15) goto(city: str, start: str, end: str, start_time: str,
    method: str):
Description: Returns a list of transportation options between
     two locations.
Parameters:
city: The city name.
start: The start point's name. Must be a location name and
    match the data exactly.
end: The end point's name. Must be a location name and match
    the data exactly.
start_time: The departure time in the format 'HH:MM'.
method: The mode of transportation, must in ['walk', 'taxi',
    'metro'].

(16) notedown(description: str, content: str):
Description: Writes the specified content to the notebook.
Parameters:
description: The description of the content.
content: The content to write.

(17) planner(query: str):
Description: Generates a plan based on the notebook content
    and query.
Parameters:
query: The query to generate a plan for. Don't worry about
    the notebook content, the planner will read it
    automatically.

(18) intercity_transport_select(start_city: str, end_city:
    str, intercity_type: str, earliest_leave_time: str = None
    ):
Description: get the intercity transportation information
    between two cities. You need to call this function at
    least twice to get the transportation information between
     two locations for going and returning.
Parameters:
start_city: The start city name.
end_city: The end city name.
intercity_type: The type of intercity transportation, must in
     ['train', 'airplane'].
```

```
earliest_leave_time: The earliest leave time in the format '
    HH:MM'.
Return: The transportation information between two cities
    sorted by leaving time.

(19) next_page():
Description: Get the next page of the latest Result history
    if it exists. Because of the length limited, all returned
     DataFrame information is split into 10 rows per page.
    You can use this function to get the next page of the
    Result history. Only DataFrame information can be split
    into multiple pages. The function should not be used too
    often, otherwise, you will soon run out of steps.
Parameters:
None

Your action will be executed in the following format: action,
     so any additional text like 'Action: ' is not allowed
    and just one line is allowed for each action.

You must finish your response within 75 steps including plan,
     otherwise the system will terminate your response. If
    you note down too often, you will soon run out of steps.
    But you can note down multiple pieces of information as a
     string WITHIN ONE CALL.

Select the transportation, dining, attractions, and
    accommodation information you need to plan your trip and
    write them in the notebook. Not EVERYTHING is needed,
    only what you need to plan the trip. For example, when
    you get ten or more accommodations, you only need to note
     down the information of the accommodation you want to
    stay in, usually one, and note it down in the notebook.
    You must not note down all the accommodations information
    . And usually, 2-4 attractions are enough for one day.

What you note down in the notebook should be a plan or plans
    for days. May be notedown(description = "Day 1(Day 1
    morning is also acceptable)", content = "At 8:00, have
    breakfast at hotel A, then go to attraction B, using
    metro(together with the cost, time, stations and other
    information). Attracion B will cost xxx yuan and xxx
    hours. Then go to restaurant C for lunch, using taxi(
    together with the cost, time, distance and other
    information). Restaurant C will cost xxx yuan.(another
    attraction is possible too as long as there is enough
    time and budget). Then... ###More details here###.")

Do not forget to note down the ###transportation information
    between locations### before planning. Intercity
    transportation information should be notedown before
    planning too.
You need to plan for each day in detail. If only one day is
    planned, accommodation is not needed. If more than one
    day is planned, accommodation is necessary. Nights in
    accommodations should be days-1.
```

```
For example, if you plan for 3 days, you need to note down 2
    nights in accommodations. Do not forget to note down the
    transportation information between locations before
    planning. Both going and returning transportation
    information should be notedown.

Call next_page() only when you need to get the next page of
    the latest Result history. Once you get the suitable
    information, you must STOP using this function. !!! Pay
    attention to function names and parameters, and the
    format of the data. You must use the correct function
    names and parameters to get the data you need. If you use
     the wrong function names or parameters, you will not get
     the correct data.!!!

The intercity transportation back to the start city must be
    notedown before planning!!!
The innercity to railway station or airport must be notedown
    before planning!!!
"""
```

