# OpenReview forum: "ChinaTravel: A Real-World Benchmark for Language Agents in Chinese Travel Planning"
_ICLR.cc/2025/Conference — ICLR 2025 Conference Withdrawn Submission_

### Official Review · Reviewer_Pfyh · 2024-10-27

**Soundness:** 2
**Presentation:** 3
**Contribution:** 3
**Rating:** 5
**Confidence:** 4

**Summary:**

This paper introduces the ChinaTravel benchmark, designed to address real-world travel planning requirements specific to China. The benchmark aggregates a large number of POIs across 10 cities, creating a rich sandbox for evaluation. Furthermore, realistic travel scenarios were developed using questionnaires, and queries were generated through either LLMs or human annotators. Agent performance is evaluated with 46 metrics encompassing feasibility, constraint satisfaction, and preference alignment. Language agents utilizing step-by-step search achieved a 51.3% success rate, highlighting both the complexity of the benchmark and the limitations of current language agents.

**Strengths:**

1. This work provides a comprehensive testbed for evaluating the travel planning abilities of current language agents in the context of Chinese travel requirements, with extensive data and high-quality, LLM-generated, or human-annotated queries.
2. The proposed benchmark is valuable for future research in travel planning for China, offering practical and substantial data entries.
3. The paper is well-structured and easy to follow, with a clear and well-presented structure and analysis.

**Weaknesses:**

1.	The success rate of search-algorithm-powered language agents falls below expectations since DFS theoretically guarantees a solution if one exists (although the search space may be vast). The reported success rate of only 51.3% is attributed to "arbitrary descriptions for defined concepts" and the "emergence of undefined concepts." Given that LLMs cannot access the entire database, they can only plan with the information provided in the environment. This limitation is more of a design issue within the system itself than a deficiency in the LLMs, as expecting agents to plan beyond their accessible information is unreasonable, even for humans. Solutions such as fuzzy matching might mitigate this issue. I wonder if the results could approach the near 100% success rate achieved in the previous approach designed for TravelPlanner[1].
2.	In the experiment, before performing DFS, all constraints are extracted, and then hard-coded rules guide the agents during the search process. This raises the question: if constraints are already extracted and hard rules still need to be manually coded, what is the added value of using language agents? Would it not be more efficient to complete the system without involving language agents at all? If the goal is to develop an autonomous agent, extracting constraints and designing code manually seems counterproductive.
3.	This work seems more like an extension of the previous TravelPlanner, adding extra constraints and adapting the setting to suit the China Travel context rather than introducing a new novel and timely benchmark the community needs now.

### Reference
[1] Hao Y, Chen Y, Zhang Y, et al. Large Language Models Can Plan Your Travels Rigorously with Formal Verification Tools[J]. arXiv preprint arXiv:2404.11891, 2024.

**Questions:**

1. In Algorithm 1, how does the validation function operate? Is it determined by the agents or by external hard rules?

---

### Official Review · Reviewer_gn6Q · 2024-11-01

**Soundness:** 2
**Presentation:** 3
**Contribution:** 2
**Rating:** 5
**Confidence:** 4

**Summary:**

This paper presents ChinaTravel, a solution targeting multi-point-of-interest (multi-POI) itineraries within selected cities in China. Notably, the authors adapt well-known in-context learning techniques and neural-symbolic approach to solve the multi-POI planning problem and evaluate their method on the ChinaTravel benchmark. A good contribution of the paper is the provision of 46 evaluation metrics that comprehensively assess:
* 23 environment constraints
* 10 hard logical constraints
* 13 preference requirements

**Strengths:**

* Originality: This is a solution targeting multi-point-of-interest (multi-POI) itineraries within selected cities in China.
* Quality: The empirical study is robust. The used technique to produce the baseline leveraged in context learning based methods and neuro-symbolic techniques, covering a good ground of applicable approaches to solve the task with LLM agents.
* Clarity: The paper is well-structured, with clear descriptions of each step in data collection and validation, providing readers with a transparent view of the methods.
* Significance: With an array of evaluation metrics, the paper provides a detailed assessment of the generated travel plans, enhancing the reliability of LLM-agent performance.

**Weaknesses:**

It is important to note that most of these constraints (environmental, logical) are extensions from existing works, enhancing their coverage aspect.

Major Comments
1. Comparative Baseline: The empirical study could benefit from a broader range of reasoning-action-based frameworks. In particular, it would be valuable to see a comparison with “Reflexion,” a reasoning-action framework considered in the TravelPlanner benchmark. This would offer a more comprehensive evaluation of ChinaTravelPlanner’s performance relative to established approaches.
2. Alternative Model Performance (Section 4.2, Line 432): In Section 4.2, the paper mentions that “...with many models failing entirely...”. Could you clarify whether additional models beyond those in Table 2 were evaluated? If so, details on their performance, especially regarding delivery, environmental, and logical pass rates, would add clarity and highlight the robustness of the proposed method.
3. Table 2 and Preference Requirements Constraints: The paper introduces preference requirements constraints, yet they are not referenced in Table 2. Including these would provide a fuller picture of constraint adherence and improve the interpretability of results across all introduced metrics.

Minor Comments (Spelling and Grammar Corrections)
* Line 140: Please revise “They integrates…” for subject-verb agreement to improve grammatical fluidity.
* Line 454: Correct “Nesy planning” to “NeSy Planning,” and add a missing period at the end of this sentence for completeness.

**Questions:**

Overall, the paper provides a meaningful contribution to multi-POI itinerary planning within an agentic framework; however, it appears to be an incremental extension of existing datasets and methods rather than a substantial innovation. The work closely parallels the objectives of the TravelPlanner paper, though adapted for a different geographical focus. The main advancements seem to be limited to alternative baseline dataset and evaluation metrics, rather than introducing new approaches or frameworks. Addressing the questions asked above could help clarify the paper’s contributions more and enhance its impact.

---

### Official Review · Reviewer_DdDN · 2024-11-03

**Soundness:** 2
**Presentation:** 3
**Contribution:** 2
**Rating:** 5
**Confidence:** 4

**Summary:**

This paper introduces ChinaTravel, a realistic travel planning benchmark designed to accommodate diverse Chinese travel requirements. Featuring a sandbox environment, ChinaTravel encompasses 10 top travel cities in China with hundreds of evaluation instances. Compared to existing benchmarks, ChinaTravel presents greater challenges due to its flexible travel requirements and a wide range of realistic metrics. Results from the STOA neural-symbolic agents, including GLM, DeepSeek, and GPT-4, illustrate the benchmark's effectiveness in distinguishing the capabilities of various large language models.

**Strengths:**

1. The paper is well-organized and easy to understand.
2. The travel planning problem presents a realistic task for evaluating the capabilities of large language models. The proposed challenging benchmark advances the field of travel planning by encouraging the development of more practical solutions.
3. As a complex task with diverse inputs and evaluation procedures, the benchmark-building process introduced in the paper is both appropriate and clearly articulated.

**Weaknesses:**

1. Compared to TravelPlanner, the differences appear to be limited, even though the authors have highlighted some distinctions in Table 1. While the most significant distinction lies in spatial coverage, other differences, such as constraints and metrics, appear less pronounced. For instance, I believe one of the most crucial metrics is the ability to guide users in articulating their requirements and to interact with them through multi-turn dialogues to refine the schedule.

2. The evaluation could be enhanced by including more results from various large language models, particularly open-source models. For example, you can consider Qwen2.5[1], LLama3.1[2] and Mistral-Small[3], in your experiments.

[1] Qwen2.5 https://qwenlm.github.io/blog/qwen2.5/

[2] Llama3.1 https://ai.meta.com/blog/meta-llama-3-1/

[3] Mistral-Small https://huggingface.co/mistralai

3. Based on the cases presented in Figure 4, the performance of large language models may be underestimated. For instance, the arbitrary descriptions in the two cases are not particularly challenging for large language models, as they simply require common sense knowledge within the travel service domain.

**Questions:**

1. Additional results from a variety of large language models, particularly open-source models, would be beneficial. For example, you can consider Qwen2.5, LLama3.1 and Mistral-Small, in your experiments.

2. Could the authors provide a more comprehensive explanation of ChinaTravel’s unique contributions? For instance, what additional evaluation metrics and travel requirements are introduced, how do these features benefit the research community, and why are they significant and challenging?

---

### Official Review · Reviewer_oexi · 2024-11-09

**Soundness:** 3
**Presentation:** 2
**Contribution:** 2
**Rating:** 5
**Confidence:** 3

**Summary:**

Real-world planning remains a challenging task for LLM agents and requires significant research effort. While neuro-symbolic reasoning demonstrates great potential in a few travel planning scenarios, real-world planning brings additional challenges. This paper proposed to create a real-world travel planning dataset with human preferences and logical constraints by considering information from 10 cities of China. The proposed approach consists of 5 main steps including manual database design, automated data generation using LLMs, automatic validation, curating requirements and constraints from human and finally creating a preference data to accommodate human expectations. A depth-first greedy planning agent is then proposed to satisfy the travel planning requirements. Experimental results demonstrate that the greedy solution typically performs better than ReAct agents, but still faces significant challenges in handling hard requirements and preferences.

**Strengths:**

Real-world travel planning is a challenging and complex task where LLM agents typically perform poorly. The author developed a reasonable size dataset with the help of user study and LLM. The real-world travel itinerary constraints and preferences are collected directly from humans which are then leveraged to generate additional synthetic information for large-scale data creation. This dataset can be useful to expand future research on LLM agentic planning. Moreover, it is demonstrated that a simple neuro-symbolic greedy method can outperform ReAct type agents.

**Weaknesses:**

Although I appreciate the effort to create large-scale travel planning dataset that will be helpful for future research, I have some concerns regarding the technical and experimental novelties of the paper:
1. The constructed dataset has only 154 queries which are generated from survey with 250 users. Therefore, it is not clear to me whether the dataset covers all possible real-world travel plan constraints and user preferences.
2. Additional constraints and requirements are generated from LLMs, but it is not described how these are validated. An automated validation method would have been great to generate large-scale data with high coverage of potential constraints.
3. The proposed NeSy planning algorithm is a simple greedy extension of neuro-symbolic planning. So, it is hard to validate the quality of the solution generated from NeSy planner. Is it possible to identify some bounds on how far the solution is from optimal preference?
4. How is the preference data handled by NeSy planner? For travel planning, it is essential to understand how LLMs deal with human preferences (e.g., optimized cost, minimal travel time and so on).

**Questions:**

1. How do you validate the accuracy of LLM generated constraints and requirement?
2. How can NeSy planner identify human preferences and come up with an optimized solution?
3. How do you guarantee the coverage of constraints and preferences in the proposed dataset?

---

### Note · Authors · 2024-11-25

I have read and agree with the venue's withdrawal policy on behalf of myself and my co-authors.